# Identification of the TSSK4 Alternative Spliceosomes and Analysis of the Function of the TSSK4 Protein in Yak (*Bos grunniens*)

**DOI:** 10.3390/ani12111380

**Published:** 2022-05-27

**Authors:** Xingdong Wang, Jie Pei, Lin Xiong, Shaoke Guo, Mengli Cao, Yandong Kang, Pengjia Bao, Xiaoyun Wu, Min Chu, Chunnian Liang, Ping Yan, Xian Guo

**Affiliations:** 1Key Laboratory of Yak Breeding Engineering of Gansu Province, Lanzhou Institute of Husbandry and Pharmaceutical Sciences, Chinese Academy of Agricultural Sciences, Lanzhou 730050, China; wxd17339929758@163.com (X.W.); peijie@caas.cn (J.P.); xionglin@caas.cn (L.X.); gsk1125@163.com (S.G.); caomengliaaa@163.com (M.C.); 82101215419@caas.cn (Y.K.); baopengjia@caas.cn (P.B.); wuxiaoyun@caas.cn (X.W.); chumin@caas.cn (M.C.); chunnian2006@163.com (C.L.); yanping@caas.cn (P.Y.); 2Key Laboratory of Animal Genetics and Breeding on Tibetan Plateau, Ministry of Agriculture and Rural Affairs, Lanzhou 730050, China

**Keywords:** yak, TSSK4, alternative spliceosomes, GST pull-down, LC–MS/MS, testis

## Abstract

**Simple Summary:**

In mammals, the testis-specific serine/threonine kinase (TSSK) is essential for spermatogenesis and male fertility. This study aimed to analyze the possible alternative spliceosomes of the TSSK4 gene in the yak testis tissue using PCR amplification and cloning techniques. A total of six alternative spliceosomes were obtained, of which there were only two alternative spliceosomes in the yak testis before sexual maturity and four alternative spliceosomes in the yak testis after sexual maturity.

**Abstract:**

In mammals, the testis-specific serine/threonine kinase (TSSK) is essential for spermatogenesis and male fertility. TSSK4 belongs to the family of the testis-specific serine/threonine-protein kinase (TSSK), with a crucial role in spermatogenesis. This study aimed to analyze the variable spliceosome of the TSSK4 gene in the yak for understanding the regulatory function of the TSSK4 spliceosome in yak testis development using PCR amplification and cloning techniques. The GST pull-down was used for pulling down the protein interacting with TSSK4, and then the protein interacting with TSSK4 was identified using LC–MS/MS. The results of the PCR amplification demonstrated multiple bands of the TSSK4 gene in the yak. The cloning and sequencing yielded a total of six alternative spliceosomes, which included only two alternative spliceosomes before sexual maturity and four alternative spliceosomes after sexual maturity. The sub-cells of the alternative spliceosomes were found to localize in the nucleus before sexual maturity and in the cytoplasm after sexual maturity. The LC–MS/MS analysis of the alternative spliceosome with the highest expression after sexual maturity yielded a total of 223 interacting proteins. The enrichment analysis of the 223 interacting proteins revealed these proteins participate in biological processes, cell composition, and molecular functions. The KEGG analysis indicated that the TSSK4-interacting protein participates in the estrogen signaling pathways, tight junctions, endoplasmic reticulum protein processing, and other signaling pathways. This study cloned the six alternative spliceosomes of the TSSK4 gene laying the foundation for studying the function of each spliceosome in the future.

## 1. Introduction

The yak (*Bos grunniens*) is a ruminant animal that was evolutionarily separated from the other ruminants about 2.2 million years ago. They can thrive in an extremely cold environment extending up to −40 °C [1], hence they are mainly distributed in the high altitudes of the Qinghai-Tibet Plateau (QTP) ranging between 2500–6000 m above the mean sea level with absolutely no frost-free period [2]. Yaks not only provide economically important products (such as meat, milk, and fur) for the local herdsmen but also serve as a means of transportation in the high mountains [3]. Therefore, the yak is considered an “all-round animal” [4], and the breeding of yak is deemed as the lifeline of the highlanders. At present, there are 20 local breeds, 2 bred breeds, and 1 wild breed of the yak as genetic resources in China. Its total number exceeds 20 million, constituting more than 95% of the total number of yaks globally [5]. Yak milk and milk products constitute the main dietary ingredients of 6.5 million people in China, also serving as an important source of income for the local families [6]. Hence, yaks are very important for Tibetan life serving as a domestic animal [7]. Despite the above advantages, yak demonstrates low reproductive efficiency with a delayed onset of puberty, seasonal reproductive patterns, low conception rate and long calving intervals [8]. The reproductive performance and breeding process in yaks are seriously restricted by low reproductive efficiency. Since testes play an important role in male reproduction, further studies on testes will help assist in increasing the reproductive efficiency of yak.

Spermatogenesis is a complicated and well-organized process that is essential for sexual reproduction, requiring strict gene expression, and involving numerous gene products [9]. At least 2% of genes in the eukaryotes specifically encode the protein kinases which phosphorylate serine (Ser), threonine (Thr), and tyrosine (Tyr) residues mainly by transferring the phosphate groups to the amino acid side chains [10]. This indicates the widespread importance of protein phosphorylation in regulating a wide range of cellular functions [11]. Protein phosphorylation is also crucial for regulating the signaling pathways and protein activity during spermatogenesis [12]. TSSK is a highly conserved group of protein kinases in various organisms [13], expressed only in the post-meiotic mammalian male germ cells. It is essential for male reproduction and plays an important role in sperm development and/or sperm function [14]. There are five reported members of the family which include TSSK1, TSSK2, TSSK3, TSSK4 (also known as TSSK5), and TSSK6 [15]. While TSSK4 is a newly-discovered member of the TSSK family, which can phosphorylate the Creb at Ser133 in vitro [16]. Meanwhile, Kueng [17] also proved TSSK4 to be related to spermatogenesis by stimulating the Cre/Creb pathway. Wei [18] reported TSSK4 to auto-phosphorylate Thr197 of the T ring, which is critical for maintaining its kinase activity. TSSK4 mechanically indicates the role of TSSK4 in the integrity and function of the sperm flagellum by phosphorylating odf2-ser76 either directly or indirectly in vivo [19]. Recent studies have demonstrated that TSSK4 induces apoptosis in vitro based on its kinase activity [20]. Mutation in the TSSK4 gene was reported in the mature sperm of infertile men in China [12]. Male mice with TSSK4 gene knockout were reported to demonstrate damaged sperm structure and decreased sperm motility, which would affect male fertility [11]. Hence, the deletion of TSSK4 can be reasonably speculated to be related to male infertility.

Alternative splicing (AS) maintains the diversity of the transcriptome and proteome and is therefore considered an essential component of regulating gene expression [21]. It can have a profound effect on gene function. In AS, multiple mature mRNAs are formed from a single pre-mRNA, which are then translated into proteins [22]. The same gene might have different isoforms which might participate in different processes, or might have directly opposite functions [23]. Reports suggest that 95% of pre-mRNAs in mammals are alternately spliced [24] since it improves the coding capacity of a single gene. Particularly, at least 15%, and perhaps as many as 50% of human genetic diseases arise from mutations either in the consensus splice site sequences or via splicing of the more variable auxiliary elements known as the exon and intron [25]. Therefore, studying the functions of the different transcripts of the same gene is necessary for the post-genomic era.

Recent years have witnessed a gradual increase in the demand for yak with the improvement of living standards. Nevertheless, yaks have low reproduction rates and long breeding cycles, usually calving once every two years or twice in three years to produce four to five calves in a lifetime [4]. The improvement in the breeding performance of yak has significantly improved the economic status of the Qinghai-Tibet Plateau and its adjacent pastoral areas. This study aimed to reveal the number of alternative spliceosomes of the TSSK4 gene in the yaks, the subcellular localization of each spliceosome, and its regulatory role in yak development. It used the predicted alternative spliceosome sequence of the yak TSSK4 gene to design primers for identifying the alternative spliceosomes and exploring the new spliceosomes of the yak TSSK4 gene. The pull-down experiment was conducted for exploring the interaction proteins of the yak TSSK4 gene variable spliceosome, providing the basic data for exploring the function of the different spliceosomes and improving the breeding abilities of yaks.

## 2. Materials and Methods

### 2.1. Ethics Statement

All the animal-related procedures conformed to the China Council on Animal Care and the Ministry of Agriculture of the People’s Republic of China guidelines. All the yak-handling procedures were approved by the Animal Care and Use Committee of the Lanzhou Institute of Husbandry and Pharmaceutical Sciences Chinese Academy of Agricultural Sciences (Permit No: SYXK-2014-0002).

### 2.2. Animals and Sample Collection

In this experiment, male yaks from four age-group were selected based on their pre-sexual and post-sexual maturities. The pre-sexually mature yaks included three 6-month-old yaks and three 18-month-old yaks, while the sexually mature yaks included three 30-month-old yaks and three 5-year-old yaks. Among them, the 6-month-old and 5-year-old yaks were slaughtered to obtain their testicular tissue, while the 18-month-old and 30-month-old yaks were castrated to obtain their testicular tissue. Before collecting the tissues, the testes were locally disinfected and post collection, the wound was sutured with a surgical needle, and penicillin and streptomycin were applied to the wound for preventing infection. The testicular tissue was quickly placed in the frozen pipe and immediately flash-frozen in liquid nitrogen before being transported to the Lanzhou Institute of Husbandry and Pharmaceutical Sciences, Chinese Academy of Agricultural Sciences for further experiments.

### 2.3. RNA Extraction and First-Strand cDNA Synthesis

The TRIzol reagent (Takara Bio Inc., Dalian, China) was used to isolate the total RNA from 12 yak testis tissues at 4 different stages. The RNA concentration and OD260/280 ratio of the samples were determined using the NanoDrop 2000 spectrophotometer (ThermoFisher Scientific, Waltham, MA, USA). The RNA concentration and OD260/280 ratio of the samples were between 500–5000 ng/mL and 1.9–2.1, respectively. The quality of the RNA was evaluated by assessing the 28S and 18S rRNA bands on 1% agarose electrophoresis gel. The qualified RNA sample was diluted to 500 ng/mL, then RNA from the testis of 3 yaks of the same age was mixed with 10 µL of each other. The cDNA was synthesized by reverse transcription of the mixed RNA using the Transcriptor First Strand cDNA synthesis kit (Takara Bio Inc., Dalian, China) and the synthesized cDNA was stored at −80 °C until further use.

### 2.4. Primer Design and Synthesis

A pair of primers (Table 1) was designed using the National Center for Biotechnology Information (NCBI) online primer design based on the predicted alternative spliceosome sequence (XM_005904917.2 and XM_005904918.1) of the yak TSSK4 gene published by the NCBI to identify the alternative spliceosome of the yak TSSK4. At the same time, it was explored whether there is a new alternative spliceosome in the yak TSSK4 gene. The primer was synthesized by Xi’an Qingke Biotechnology Co., Ltd.

### 2.5. PCR Amplification, Cloning and Sequencing of the TSSK4 Gene

After completing the PCR reaction (reaction system for Table 2, and reaction conditions for Table 3), the PCR products were stored in a 4 °C refrigerator. About 2.5 μL of PCR amplification product was detected by 4% agarose gel electrophoresis and observed under an automatic digital gel imaging system (Tanon, Shanghai, China) and photographed and stored.

The PCR product was recovered according to the instructions of the Universal DNA Purification Kit (Tiangen, Beijing, China), then the pMD19-T cloning vector and the recovered product were placed in a water bath at 16 °C overnight to form the recombinant plasmid. The ligation product was transformed into a semi-melted competent cell BL21, in the ice bath for 20 min followed by a heat shock at 42 °C for 1 min and then incubated on ice for 3 min; 600 μL of LB (Amp+) liquid medium was added at 37 °C for 50 min at constant temperature (220 r/min); 200 µL of the solution was taken and plated onto a petri dish, incubated in the incubator at 37 °C for 12 h; each plate was picked for four bacteria, and each tube containing 5 mL LB (Amp+) was incubated in a shaker at 37 °C for 12 h; the bacterial liquid was verified by PCR and the positive bacterial liquid was selected for sequencing by Xi’an Qingke Biotechnology Co., Ltd.

### 2.6. Prokaryotic Expression and Purification of the TSSK4 Protein

The alternative spliceosome with the highest expression level in the testis after sexual maturation was selected and evaluated theoretically for the hydrophilicity, signal peptide, transmembrane domain, and basic structure of its protein sequence. After the evaluation, the plasmid was constructed by gene fishing and subcloned into the expression vector, pGEX-4T-1 (Sangon Biotech, Shanghai, China). The cells were further transformed into the Rosetta (DE3) *Escherichia*
*coli* receptor cells, after which it was cultured, induced and expressed, collected, and purified to obtain 1 mg, purity >80%, freeze-dried recombinant TSSK4 protein.

The target protein was verified using Western blot (see Appendix A for the detailed process). The samples were first processed and boiled in a water bath at 95–100 °C for 5 min. The separating gel and stacking gel (Appendix A) were prepared, the samples were added to the spotting well for electrophoresis, transferred onto the membrane, and the transferred membrane was incubated in the blocking solution for 1 h at room temperature. The blocking solution was removed and the membranes were incubated in the primary antibody (GST antibody; Sangon Biotech, Shanghai, China; 1: 50,000 dilution) overnight at 4 °C. The primary antibody was recovered and the membrane was washed with TBST, the secondary antibody (rabbit anti-GST antibody; Sangon Biotech, Shanghai, China; 1:10,000) was added followed by incubation at room temperature for 30 min, followed by exposure in the darkroom, after develop and fixation.

### 2.7. GST Pull-Down Verifying the Interaction Protein of the TSSK4 Protein

The control group was mixed with 500 μg GST and 200 μL 50% glutathione-agarose resin homogenate. The experimental group was mixed with 500 μg GST-TSSK4 and 200 μL 50% glutathione-agarose resin homogenate and gently shaken at 4 °C for 2 h. The pellet was centrifuged at 2500× *g* rpm for 3 min and the supernatant was discarded. The precipitate was washed by adding 1 mL PBST, and mixed upside down for removing the mixed proteins that were not bound to the resin. The mixture was centrifuged at 2500× *g* rpm for 3 min at 4 °C and the supernatant was discarded, and steps 2 and 3 were repeated three times. The whole protein was extracted from the testis tissue of the 5-year-old yaks using the whole protein extraction kit (Solarbio, Beijing, China). After passing through the protein test, 5 mg of total protein was added to the control group and the experimental group and incubated overnight at 4 °C. The protein mixture was centrifuged at 2500× *g* rpm for 3 min at 4 °C and the supernatant was discarded. The pellet was then washed with 1 mL of pre-cooled PBST, centrifuged at 2500× *g* rpm for 3 min at 4 °C, the supernatant was discarded, and the washing was repeated three times. The pellet was then treated with 80 μL of RIPA Buffer cell lysate and 20 μL 6× Loading Buffer, mixed well, boiled in boiling water for 5–10 min, centrifuged at 12,000× *g* rpm for 5 min, and the supernatant was collected. After the supernatant was separated using SDS-PAGE electrophoresis, the protein band was transferred to the PVDF membrane using the wet transfer method for Western blot detection (see Appendix A for the detailed process).

### 2.8. Enzymolysis in the Protein Gel

The target strip was cut into cubes of 0.5–0.7 mm with a sterilized knife. After decolorization with the test stain/silver staining decolorizing solution, the glue block was cleaned with 500 μL acetonitrile solution thrice until it turned completely white. Then 500 µL 10 mM DTT was added to the water bath at 56 °C for 30 min followed by centrifugation at low speed. Subsequently, 500 µL decolorization solution was added and mixed well at room temperature (20–25 °C) for 5–10 min, centrifuged, and the supernatant was discarded. To this 500 µL 55 mM IAM was added and placed in a dark room at room temperature for 30 min before centrifugation; 500 μL decolorization solution was added, mixed well at room temperature for 5–10 min, centrifuged, the supernatant was discarded. A total of 500 μL acetonitrile was added to the above solution until the colloidal particles turned white, and vacuum dried for 5 min; 0.01 µg/µL trypsin was added according to the volume of the gel, and an appropriate amount of NH_4_HCO_3_, pH 8.0, was added in the ice bath for 30 min and enzymolized overnight at 37 °C. After enzymatic hydrolysis, 300 µL extract was added and the ultrasound was performed for 10 min and centrifuged at low speed to collect the supernatant. This step was repeated twice, and the extract was obtained and combined and kept in a vacuum.

### 2.9. Zip-Tip Desalting

The sample obtained in the previous step was dissolved in 10–20 μL 0.2% TFA solution, centrifuged at 10,000 rpm for 20 min, wetted Ziptip with infiltration solution for 15 times, and balanced Ziptip with equilibrium solution for 10 times. The sample solution was inhaled for 10 cycles and blown 8 times with the rinsed solution. Then 50 μL of eluent was added into a clean EP tube, the eluent was blown several times, and the sample was drained.

### 2.10. LC–MS/MS Analysis

The polypeptide samples were diluted to 1 μg/μL using the buffer, the sample volume was set to 5 μL and the scanning mode was collected for 60 min. Peptides with mass to charge ratios of 350–1200 were scanned. The mass spectrometry data were collected using the Triple TOF 5600 + Liquid mass spectrometry system (ABSCIEX, Redwood City, CA, USA). The peptide samples were dissolved in 2% acetonitrile/0.1% formic acid and analyzed using a Triple TOF 5600 Plus mass spectrometer coupled with the Eksigent nanoLC System (ABSCIEX, Redwood City, CA, USA). The polypeptide solution was added to a C18 capture column (3 μm, 350 μm × 0.5 mm, AB Sciex, Redwood City, CA, USA) at a time gradient of 60 min. The gradient elution was performed on a C18 column (3 μm, 75 µm × 150 mm, Welch Materials, Inc., Hefei, China) at a flow rate of 300 nL/min. The two mobile phases were buffer A (2% acetonitrile/0.1% formic acid/98% H_2_O) and buffer B (98% acetonitrile/0.1% formic acid/2% H_2_O). For IDA (information-dependent acquisition), the primary mass spectrometry was performed at 250 ms ion accumulation time, and secondary mass spectrometry of 30 precursor ions was collected at 50 ms ion accumulation time. The MS1 spectra were collected in the range of 350–1200 *m*/*z*, and the MS2 spectra were collected in the range of 100–1500 *m*/*z*. The dynamic removal time of the precursor ions was set to 15 s.

### 2.11. Data Analysis

The original MS/MS data of the mass spectrometer were transferred to the ProteinPilot (https://sciex.com.cn/products/software/proteinpilot-software, accessed on 23 October 2021, version 4.5, SCIEX, Redwood City, CA, USA). For protein identification, the Paragon algorithm in ProteinPilot was used to search the uniprot-Bos_mutus-uniprot-organism_72004.fasta database. The parameters were set as follows: the instrument was TripleTOF 5600, and the cysteine was modified with iodoacetamide; the biological modification was selected as the ID focus. For identifying the protein certain filtering criteria were selected, and peptides with an unused score >1.3 (with a confidence level of 95% or more) were considered credible peptides, and the proteins containing at least 1 unique peptide were retained. The functional enrichment analysis of interacting genes with TSSK4 was performed using the g: Profile online website (https://biit.cs.ut.ee/gprofiler/gost, accessed on 13 December 2021) and KOBAS online website (http://kobas.cbi.pku.edu.cn/, accessed on 13 December 2021).

## 3. Results

### 3.1. Identification and Subcellular Localization of the Alternative Spliceosomes of the Yak Tssk4

This experiment used a total of 12 yaks in four age stages: 6 m (6-month-old yak), 18 m (18-month-old yak), 30 m (30-month-old yak), and 5 Y (5-year-old yak). The RNA samples were isolated from each stage sample and pooled, respectively. The PCR amplification using specific primer T1 designed by NCBI, followed by a 4% agarose gel electrophoresis revealed the 6-month-old and 18-month-old yaks to possess bright and clear bands above 2000 bp, while the 30-month-old and 5-year-old yaks were found to possess multiple bright bands aggregated over 1000 bp (Figure 1a). Therefore, multiple alternative spliceosomes can be speculated in the yak TSSK4 gene. The PCR product was cloned to further explore the alternative spliceosome of the TSSK4 gene in yak and 50 monoclonals were selected from samples in each group for sequencing. A total of six alternative spliceosomes of the TSSK4 gene were found, with GenBank accession numbers ranging between MT950338–MT950343. Moreover, there were no shared alternative spliceosomes before and after the sexual maturation (Table 4).

The protein subcellular localization online website PSORT II (https://psort.hgc.jp/form2.html, accessed on 13 December 2021) was used to predict the subcellular localization of the six alternative spliceosomes of the yak TSSK4, and the two alternative spliceosomes (MT950341 and MT950342) before sexual maturity were found to localize in the nucleus, while the four alternative spliceosomes (MT950338–MT950340, MT950343) were found to localize subcellular in the cytoplasm after sexual maturation.

### 3.2. Prokaryotic Expression and Purification of the TSSK4 Protein

To explore the interacting proteins of the alternative spliceosomes, the alternative spliceosome (MT950343) with the highest expression level after sexual maturation was selected. The recombinant plasmid was verified through enzyme digestion and yielded two bands in the gel diagram, one of which was the same size as the spliceosome of MT950343, and the other was the same size as the vector pGEX-4T-1 (Figure 1b), which indicated that the recombinant plasmid was successfully purified. The purified fusion protein was then analyzed using SDS-PAGE with obvious bands near the theoretical molecular weight (65 KD) (Figure 1c), which preliminarily confirmed the successful purification of the fusion protein. The purified protein was further determined as the target protein using the TMB color kit based on color development, following the steps of Western Blot. The results showed that obvious bands appeared in the corresponding position (Figure 1d), indicating it to be a target protein.

### 3.3. Identification of the TSSK4-Interacting Proteins by GST Pull-Down Combined with the LC–MS/MS Mass Spectrometry

The GST-glutathione-agarose resin homogenate and GST-TSSK4-glutathione-agarose resin homogenate were used to investigate the interaction proteins of the yak TSSK4 protein. The silver staining yielded only one band in the GST-TSSK4 lane, a small number of bands in the control lane (GST-glutathione-agarose resin homogenate), and a large number of bands in the experimental lane (GST-TSSK4-glutathione-agarose resin homogenate) (Figure 2a). The Western blot detected positive reactions in the control group, experimental group, and GST-TSSK4 lane (Figure 2b), and the band size of the GST-TSSK4 lane and experimental group lane was the same, and the band size of the control group corroborated with the expected experiment (GST band size). This confirmed the successful pull-down of the proteins interacting with the GST and GST-TSSK4.

The number of secondary spectrograms produced after LC–MS/MS, using GST and GST_TSSK4 mass spectrometry of the samples was 4921 and 11459, respectively, and the number of secondary spectrograms was analyzed to be 720 and 3828, respectively (Table 5). A total of 246 proteins were identified using GST and GST_TSSK4, among which 18 proteins were simultaneously identified for two samples. The specific proteins identified by GST and GST_TSSK4 were 5 and 223, respectively (Figure 2c), while the total number of proteins identified by the GST and GST_TSSK4 protein samples were 23 and 241, respectively (Appendix A). The relevant information on some proteins with high scores is summarized in Table 6.

### 3.4. Enrichment Analysis of the TSSK4-Interacting Proteins

The 223 interacting proteins screened out were subjected to the GO enrichment analysis, and these proteins were concluded to have a role in biological processes, cell composition, and molecular functions. (1) Molecular function: structural constituent of the cytoskeleton, act in binding the zinc ion; (2) Cell component: Golgi apparatus, endoplasmic reticulum, keratin filament; (3) Biological process: cell differentiation, negative regulation of the apoptotic process, response to estradiol (Figure 3a). The KEGG analysis revealed that TSSK4 interaction protein participates in the estrogen signaling pathway, tight junction, protein processing in the endoplasmic reticulum, and other signaling pathways (Figure 3b).

## 4. Discussion

The genomic diversity is increased through the generation of multiple mRNA isoforms from a single gene by the alternative splicing (AS) [26]. More than 95% of the human genes undergo AS after transcription [27], causing multiple protein subtypes to be generated from a primary transcript. This transcript is one of the main drivers of the proteome diversity in human cells [28]. AS not only determines the cell fate and embryo development [29], but also regulates the tissue or developmental specific procedures, and their abnormal expression is involved in triggering many diseases [30,31]. TSSK is highly expressed in the testis and is responsible for regulating many protein activities related to spermatogenesis [9]. TSSK4 belongs to the TSSK family, expressed only in the testis, with an important role in male spermatogenesis [15].

To date, a large number of TSSK genes and even some alternative splicing bodies of the TSSK genes have been reported in various animals. However, many of these genes and their alternative splices are only predicted based on sequence similarity [9]. For example, only TSSK6 was tested experimentally among the TSSK genes of sheep and goats, and the other members were predicted by calculation [9]. Therefore, the existence of these genes and their alternative spliceosomes need to be verified through experimental methods.

This experiment used two predicted sequences (XM005904918.1 and XM005904917.2) of NCBI as templates to design the primer, and the testicular tissue of the yak was used as the research object. The PCR amplification, cloning, and sequencing were carried out, and finally, six alternative spliceosomes of the TSSK4 gene were obtained. The comparison analyses revealed that only the sequencing result MT950339.1 was consistent with the predicted sequence XM005904918.1, while the other five sequencing results were not consistent with the predicted sequence. There were two alternative spliceosomes identified in the TSSK4 gene of the tree shrew [32], three identified in the TSSK4 gene of humans and pigs [9], and four identified in the TSSK4 gene of mouse testis [18]. The TSSK4 gene of the yak testes identified six alternative spliceosomes. The analysis revealed the TSSK4 spliceosomes of tree shrews, humans, pigs, and mice to originate from the sexually mature individuals, while the TSSK4 of yak were found to not only originate from the testicular tissue after sexual maturity but also from the testicular tissues before sexual maturity. The length of the alternative splice variants of the yak TSSK4 gene was reported to be relatively close, therefore attempts were made to isolate them from the gel, but the final separation failed. Therefore, the splice variants were isolated by cloning and picking the monoclonal bacteria. To isolate all the splicing variants as much as possible, 50 monoclonal bacteria were selected from each period. After sequencing, the monoclonal results obtained at each stage were sorted out and the results obtained from the samples of the two periods after sexual maturity were identified to be similar. The results obtained from the samples of the two periods before sexual maturity were also found to be similar, which indicated that the experimental results were accurate. Comparing the result of sequencing and predicting the sequence revealed that there could be either of these probabilities: (1) the predicted sequence (XM005904917.2) might not exist in the yak testis; (2) since the predicted sequence expressed in extremely low levels in the yak testis, the monoclonal bacteria liquid might not have been selected. Cloning and sequencing results revealed the yak TSSK4 gene to express completely different splice variants before and after sexual maturity. Bioinformatics analyses revealed the splice variant of TSSK4 to localize in the nucleus before sexual maturity, and in the cytoplasm after attaining sexual maturity, establishing completely inconsistent subcellular localization. A previous study [32] on tree shrews after sexual maturity revealed the TSSK protein to localize in the cytoplasm, consistently with the subcellular localization of the TSSK protein in previous reports in mice [33]. After synthesis, the protein can only perform its function correctly provided it is transported to the corresponding subcellular location. Different sub-cellular organelles regulate different functions in the cells, and the mislocalization of the proteins triggers a series of diseases [34,35].

The GO enrichment analysis revealed the proteins interacting with TSSK4 to involve in the functions such as the cytoskeletal structural components, Golgi apparatus, and endoplasmic reticulum. The cytoskeleton is a network system comprising protein components distributed throughout the eukaryotic cells. It is mainly involved in processes such as the replication, expression, and transmission of the genetic material of the cell, forming the material basis for the cells to function normally [36]. The parallel tubular structures, microtubules comprise α-tubulin and β-tubulin, with important roles in cell division and movement, and the maintenance of cell morphology and structure [37]. Hence, the normal functions and structures of the microtubules determine the stability of the structure of the spermatogenic epithelial junction.

The proteins identified by LC–MS/MS were enriched in the estrogen signaling pathway. Estrogen was first isolated nearly 90 years ago accounting for possible roles pertaining to female reproduction. With expanding research, estrogen was also found to be produced in men [38]. The estrogen receptors (ER) are abundantly expressed in the Mullerian and Wolfe ducts of the developing male reproductive tract and the urogenital sinus, as well as in the reproductive organs of the late fetuses and newborns derived from the Mullerian and Wolfe ducts [39]. Cloning of human ER [40] and development of ER antibodies [41] has recognized the efferent tubule as the main target of male estrogen [42]. The reproductive phenotypes of the Esr1 KO male rats are similar to that of the Esr1 KO mice, which include infertility, decreased testicular weight, and sperm in the epididymal tail [43]. However, estrogen was recently identified to play a role in spermatogenesis in the testes and epididymis [44,45,46] and to maintain certain aspects of epididymal physiology [47,48]. Estrogens, ER, and the whole estrogen signaling pathway are important for regulating the male fertility, development, and function of the efferent ducts and prostate, and the flow of the adult sperm from the testis to the epididymis [49].

Although the TSSK family proteins are essential for testicular development and spermatogenesis, not much research has currently been performed on the TSSK family proteins in the yak. The mechanism by which the TSSK protein regulates testicular development and sperm production in yaks remains unknown. This study designed primers based on the predicted sequences of the yak TSSK4 gene for amplifying the TSSK4 gene sequences in yaks and amplified six alternative spliceosomes. Due to the high degree of similarity in the six alternative splices, it was impossible to design primers and prepare antibodies for the quantitative analysis. Therefore, a large-scale selection of the monoclonal bacteria is required to measure the proportion of each alternative spliceosome. Functional studies on the six alternative splice variants of TSSK4 were predicted mainly through bioinformatics methods. At the same time, the prokaryotic expression was only assessed in the variable spliceosome with the highest expression level after sexual maturity, after studying their interacting proteins using LC–MS/MS. However, the different spliceosomes of the same gene might have different functions. Therefore, the other five alternative spliceosomes of the TSSK4 gene should be expressed later for studying their functions. Studying the various alternative spliceosomes of the TSSK4 gene in yaks usher a hope to further understand the regulatory effects of the different alternative spliceosomes in developing and producing sperm in yaks.

## 5. Conclusions

This experiment identified six alternative spliceosomes of the TSSK4 gene at four age stages of the yak using PCR amplification and cloning technology. Among them, only two alternative spliceosomes were found in the yak before sexual maturity (6 months old and 18 months old). After sexual maturity (30 months old and 5 years old), four alternative spliceosomes were amplified in the yak.

## Figures and Tables

**Figure 1 animals-12-01380-f001:**
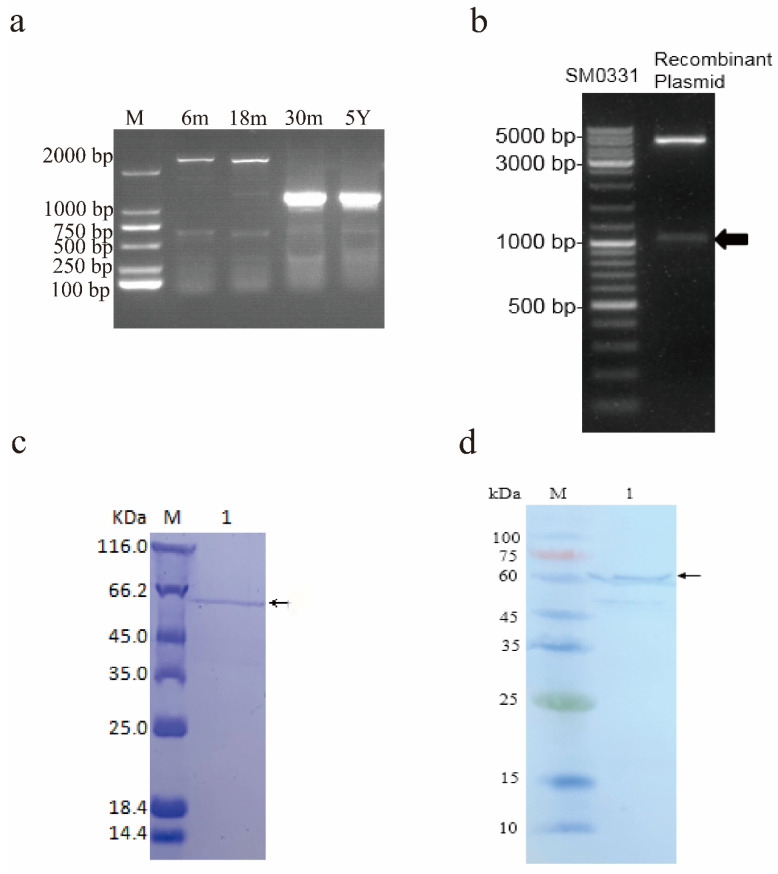
Identification and protein purification of the yak TSSK4 alternative spliceosomes. (**a**) Agarose gel electrophoresis of the PCR product of the TSSK4 gene. M represents 2000 bp DNA marker, while 6 m, 18 m, 30 m, and 5 Y represent 6-month-old yak, 18-month-old yak, 30-month-old yak, and 5-year-old yak, respectively. (**b**) Enzyme digestion of the recombinant plasmid. Marker: SM0331. (**c**) Validation of the fusion protein. M: Protein marker; 1: Fusion of the target protein. (**d**) Western blot analysis of the purified protein. M: Protein marker; 1: Fusion of the target protein.

**Figure 2 animals-12-01380-f002:**
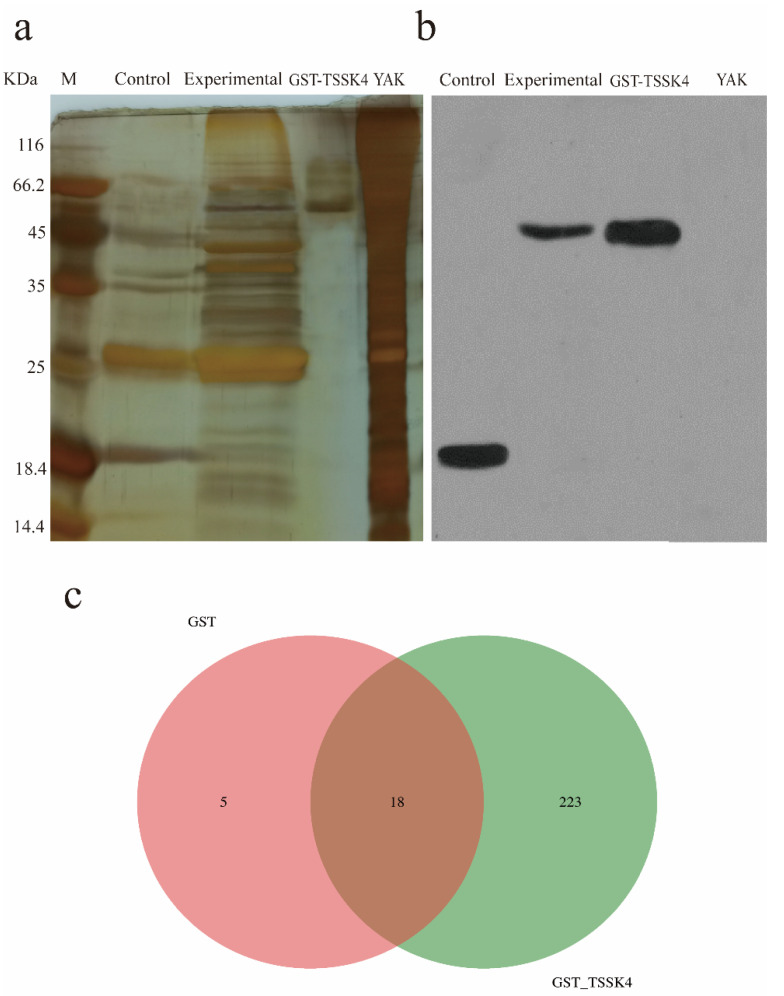
The results of the GST pull-down and LC–MS/MS mass spectrometry identification. (**a**) The results of the silver staining. Note: The control group and the experimental group were loaded with a 20 µL sample, GST-TSSK4, yak quantitative 2 µg. (**b**) The results of the GST antibody test. The control and the experimental groups were loaded with a 20 µL sample, GST-TSSK4, yak loaded with 50 ng, GST antibody was diluted at 1:50,000, and hypersensitivity was exposed for 2 min. (**c**) IP: Venn diagram of the IgG differential proteins.

**Figure 3 animals-12-01380-f003:**
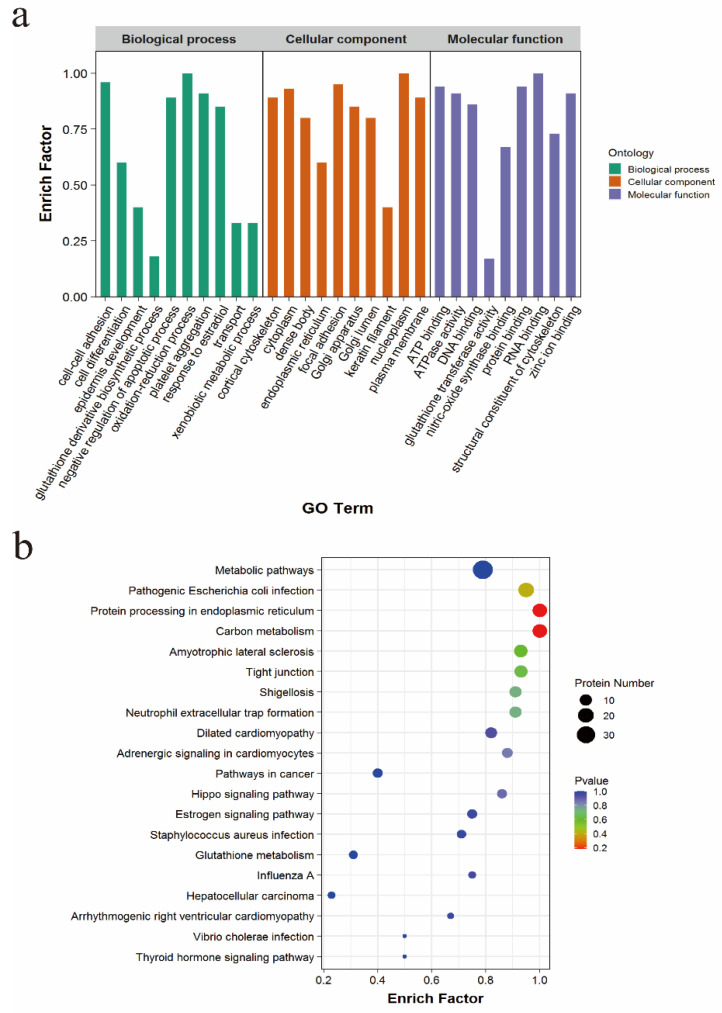
Enrichment analysis. (**a**) GO enrichment analysis of TSSK4-interacting protein. (**b**) KEGG enrichment analysis of the TSSK4-interacting protein.

**Table 1 animals-12-01380-t001:** Primer sequences information.

Names	Primer Sequences (5’→3’)	Product Sizes (bp)	Annealing (Tm, °C)	Notes
T1	TTGCATAGGCAAGCTTTTGG	1216	60	Clone
	ATTTTTATCCCCAGACCCTCC			

**Table 2 animals-12-01380-t002:** The composition of PCR.

Element	Concentration	Direction (µL)
2 × Taq Mix	2×	12.5
Forward Primer	100 pmol/L	1
Reverse Primer	100 pmol/L	1
cDNA	500 ng	1
ddH^2^O		9.5
Total		25

**Table 3 animals-12-01380-t003:** The parameters for PCR response.

Project	Temperature/°C	Time	Cycles
Pre-denaturation	94	2	×1 cycles
Denaturation	94	1	×35 cycles
Annealing	60	1	
Extension	72	2	
Extension	72	5	×1 cycles
Save	4	∞	×1 cycles

**Table 4 animals-12-01380-t004:** The number of alternative spliceosomes at the different stages of the yak TSSK4 gene.

Group	MT950343	MT950338	MT950339	MT950340	MT950341	MT950342
6 m	0	0	0	0	0	50
18 m	0	0	0	0	2	48
30 m	33	2	12	3	0	0
5 Y	33	1	12	4	0	0

**Table 5 animals-12-01380-t005:** Summary table of the protein identification information statistics.

Sample	The Score Figure Number	Identify the Number of Spectrograms	Spectral Resolution (%)	Identify the Number of Peptides *	Identifying Protein Number	Unique-2 **
GST	4921	720	14.63	180	23	21
GST_TSSK4	11459	3828	33.41	1285	241	168

Note: * indicates at least 95% confidence, ** indicates the number of identified proteins containing at least 2 unique peptides.

**Table 6 animals-12-01380-t006:** Protein-related information sheet.

Protein ID	Coverage (%)	Mass (Da)	Unique Peptide
tr|A0A6B0QPS2|A0A6B0QPS2_9CETA	76.44	26,849.8	27
tr|L8HWB9|L8HWB9_9CETA	61.34	22,418.0	12
tr|A0A6B0SBF2|A0A6B0SBF2_9CETA	48.80	41,736.4	6
tr|L8I5A7|L8I5A7_9CETA	51.43	23,580.9	14
tr|A0A6B0QRL7|A0A6B0QRL7_9CETA	62.39	25,634.6	8
GST-TSSK4	10.93	65,318.1	10
tr|L8HP74|L8HP74_9CETA	17.82	51,866.2	6
tr|L8IU57|L8IU57_9CETA	45.05	26,112.9	8
tr|L8HXW6|L8HXW6_9CETA	8.01	66,485.0	2
tr|A0A6B0S215|A0A6B0S215_9CETA	2.56	240,437.2	6

## Data Availability

The datasets generated and analyzed during the current study are available in the GenBank repository, accession number: MT950338–MT950343. The data supporting the conclusions of this study are available within the Appendix A.

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
