# Peer review of "Identification of the TSSK4 Alternative Spliceosomes and Analysis of the Function of the TSSK4 Protein in Yak (Bos grunniens)"

_animals, 2022, doi:10.3390/ani12111380_

Round 1

Reviewer 1 Report

The present work is verry interesting and well-written, however, some editorial corrections are necessary. The work focuses on yak, which as the Authors underlined, is very important animal from the economic point of view. The obtained results are novel and interesting, however there is some gaps in the text that has to be filled. For more, I have some concerns about the study design.  

Specific comments and questions:

2.2. Animals and Sample Collection

In my opinion the number of animals n=3 per group is too small. In this type of studies, the minimal number of individuals should be 4 or 5 per group.

2.3. RNA extraction, and first-strand cDNA synthesis

Was the RNA extracted from the whole testes homogenate or from any specific region, structure? It is important since in the animals from different age groups the ratio between different structures changes. As long as the different structures express different set of proteins and mRNA, the difference in their ratio may influence the final results.

My biggest concern about the study design is that the Authors conducted their experiments on the probes with mixed cDNA from all individuals from the group. In my opinion the experiments should be conducted for each individual from the group separately and then the results should be statistically compared. In the case of the Authors’ approach, we cannot say that the experiment was run for the n=3 per group from its definition.

2.4. Primer design and synthesis

Did the Authors conduct the validation experiment for the designed primers? What was the efficiency of primers?

Please transfer the sequence symbols and reaction conditions to the table.

2.5. PCR amplification, cloning, and sequencing of the tssk4 gene

Please replace the volume of the primers with their concentration.

Did the Authors conducted the sequencing procedure of the obtained product to confirm its specificity?

2.6. Prokaryotic expression and purification of the TSSK4 protein

Please attach the more detailed Western blot protocol (electrophoresis protocol, transfer method, membrane blocking protocol, antibodies dilutions and supplier etc.)

Please describe the Western blot protocol in the separate paragraph.

Line 174: Lack of capital letter at the beginning of the sentence.

Line 175: Lack of the dot at the end of the sentence

Line 185: Lack of space.

2.8. Enzymolysis in the protein gel

“To this 500 µL 55 mM IAM was added, and placed in a dark room at room temperature for 30 min before centrifugation; 500 µL decolorization solution was added, mixed well at room temperature for 5–10 min, centrifuged, the supernatant was discarded; 500 µL acetonitrile was added until the colloidal particles turned white, and vacuum dried for 5 min; 0.01 µg/µL trypsin was added according to the volume of the gel, and an appropriate amount of NH4HCO3, pH 8.0, was added in the ice bath for 30 min, and enzymolized overnight at 37°C.” – This sentence is too long.

2.10. LC-MS/MS analysis

How many probes for each group was descended for the LC-MS experiments?

There is total lack of the validation step of the obtained LC-MS data. The obtained results should be validated with Western blot for 4 to 5 chosen proteins to confirm the correctness of the method and results. Please attach the description of the validation experiment and the obtained results.

2.11. Data Analysis

Lines 234-235: Unnecessary space

Did the Authors base on the P-value or on the FDR? In this case the FDR values are more important and commonly used for the high throughput data sets obtained from the transcriptomic or proteomic experiments.

3.1. Identification and subcellular localization of the alternative spliceosomes of the yak Tssk4

It would be very beneficial for the present studies to confirm the predicted localization of the spliceosomes with the use of cellular in situ hybridization experiments.

3.3. Identification of the TSSK4 interacting proteins by GST pull-down combined with the LC-MS/MS mass spectrometry

What was the Fold Change (FC) and p-value cutoff level?

  1. Discussion

The discussion is very descriptive and should be remodeled. I suggest the authors to focus more on the individual proteins revealed in the LC-MS analysis than on the GO results.

Line 374, after (2): The capital letter is unnecessary.

Lines 386-387: The part of the discussion above, as well as the presented results did not support this thesis.

Lines 398-412: This part of discussion is only the description of the estrogen role in the male reproductive system and is not addressed to the obtained results and its potential impact on the physiology.

Author Response

Dear Reviewer:

Thank you for your time concerning our manuscript entitled “Identification of the Tssk4 alternative spliceosomes and analy-sis of the function of the TSSK4 protein in yaks” (written by Xingdong Wang et al.), which we wish to be considered for publication in “Animals”. The following is a point-by-point response to your comments.

2.2. Animals and Sample Collection

In my opinion the number of animals n=3 per group is too small. In this type of studies, the minimal number of individuals should be 4 or 5 per group.

Response: Thank you for your advice. In order to better explore the alternative spliceosome of the Tssk4 gene in yak at various stages, more samples should be collected in each stage. At the same time, in order to eliminate the influence of the yak family, the selected yaks are not related. The kinship of yaks needs to be traced before the experiment, so there were certain difficulties in the collection of experimental samples. In addition, some herdsmen do not agree to castration sampling. Therefore, considering various factors, three yaks were collected in each period.

2.3. RNA extraction, and first-strand cDNA synthesis

Was the RNA extracted from the whole testes homogenate or from any specific region, structure? It is important since in the animals from different age groups the ratio between different structures changes. As long as the different structures express different set of proteins and mRNA, the difference in their ratio may influence the final results.

Response: Thank you for your advice. RNA was extracted from the whole testicular tissue of yaks. At the beginning of the experimental design, we considered that there may be differences in RNA in specific regions and structures, so RNA was extracted from the whole testis tissue.

My biggest concern about the study design is that the Authors conducted their experiments on the probes with mixed cDNA from all individuals from the group. In my opinion the experiments should be conducted for each individual from the group separately and then the results should be statistically compared. In the case of the Authors’ approach, we cannot say that the experiment was run for the n=3 per group from its definition.

Response: Thank you for your advice. Before the experiment, we conducted a pre-experiment, using primers to amplify the tssk4 gene in the testes of 3 yaks in each period, and identified by gel electrophoresis (3%) that the electrophoresis bands of the 3 yaks in each period were consistent. The purpose of the experiment was to amplify all the alternative spliceosomes of the tssk4 gene in the yak at this stage as much as possible, so the effect of using mixed samples is the same as that of single individual amplification.

2.4. Primer design and synthesis

Did the Authors conduct the validation experiment for the designed primers? What was the efficiency of primers?

Response: Thank you for your advice. The amplification of Yak tssk4 gene was also a validation experiment of primers. RT-qPCR was not involved in the experiment, therefore, there was no need to verify primer efficiency.

Please transfer the sequence symbols and reaction conditions to the table.

Response: Thank you for your advice. The reaction conditions and reaction procedures etc. have been transferred to Tables 2 and 3 as required.

Table2. The composition of PCR

Element

concentration

Direction (µl)

2×Taq Mix

12.5

Forward Primer

100pmol/L

1

Reverse Primer

100pmol/L

1

cDNA

500ng

1

ddH2O

9.5

Total

25

Table3. The parameters for PCR response

Project

Temperature/℃

Time

Cycles

Pre-denaturation

94

2

×1 cycles

Denaturation

94

1

×35 cycles

Annealing

60

1

Extension

72

2

Extension

72

5

×1 cycles

Save

4

×1 cycles

2.5. PCR amplification, cloning, and sequencing of the tssk4 gene

Please replace the volume of the primers with their concentration.

Response: Thank you for your advice. Primers concentrations have been indicated in Table 2

Did the Authors conducted the sequencing procedure of the obtained product to confirm its specificity?

Response: Thank you for your advice. The obtained products were confirmed by Sanger sequencing

2.6. Prokaryotic expression and purification of the TSSK4 protein

Please attach the more detailed Western blot protocol (electrophoresis protocol, transfer method, membrane blocking protocol, antibodies dilutions and supplier etc.)

Response: Thank you for your advice. Western Blot Protocol (electrophoresis protocol, transfer method, membrane blocking protocol, antibodies dilutions and supplier etc.) has been added into the supplementary materials as required.

Please describe the Western blot protocol in the separate paragraph.

Response: Thank you for your advice. The description of the Western blot Protocol was added in lines 178-189 of the paper. For details, see supplementary materials.

Line 174: Lack of capital letter at the beginning of the sentence.

Response: Thank you for your advice. “the experimental group was mixed with 500 μg GST-TSSK4 and 200 μL 50% glutathione-agarose resin homogenate, and gently shaken at 4°C for 2 h,” has been replaced by “The experimental group was mixed with 500 μg GST-TSSK4 and 200 μL 50% glutathione-agarose resin homogenate, and gently shaken at 4°C for 2 h.”

Line 175: Lack of the dot at the end of the sentence

Response: Thank you for your advice. “the experimental group was mixed with 500 μg GST-TSSK4 and 200 μL 50% glutathione-agarose resin homogenate, and gently shaken at 4°C for 2 h,” has been replaced by “The experimental group was mixed with 500 μg GST-TSSK4 and 200 μL 50% glutathione-agarose resin homogenate, and gently shaken at 4°C for 2 h.”

Line 185: Lack of space.

Response: Thank you for your advice. space has been added to the middle of at 2500.

2.8. Enzymolysis in the protein gel

“To this 500 µL 55 mM IAM was added, and placed in a dark room at room temperature for 30 min before centrifugation; 500 µL decolorization solution was added, mixed well at room temperature for 5–10 min, centrifuged, the supernatant was discarded; 500 µL acetonitrile was added until the colloidal particles turned white, and vacuum dried for 5 min; 0.01 µg/µL trypsin was added according to the volume of the gel, and an appropriate amount of NH4HCO3, pH 8.0, was added in the ice bath for 30 min, and enzymolized overnight at 37°C.” – This sentence is too long.

Response: Thank you for your advice. “To this 500 µL 55 mM IAM was added, and placed in a dark room at room temperature for 30 min before centrifugation; 500 µL decolorization solution was added, mixed well at room temperature for 5–10 min, centrifuged, the supernatant was discarded; 500 µL acetonitrile was added until the colloidal particles turned white, and vacuum dried for 5 min; 0.01 µg/µL trypsin was added according to the volume of the gel, and an appropriate amount of NH4HCO3, pH 8.0, was added in the ice bath for 30 min, and enzymolized overnight at 37°C.” has been replaced by “To this 500 µL 55 mM IAM was added, and placed in a dark room at room temperature for 30 min before centrifugation; 500 μL decolorization solution was added, mixed well at room temperature for 5–10 min, centrifuged, the supernatant was discarded. 500 μL acetonitrile was added to the above solution until the colloidal particles turned white, and vacuum dried for 5 min; 0.01 µg/µL trypsin was added according to the volume of the gel, and an appropriate amount of NH4HCO3, pH 8.0, was added in the ice bath for 30 min and enzymolized overnight at 37°C.”

2.10. LC-MS/MS analysis

How many probes for each group was descended for the LC-MS experiments?

Response: Thank you for your advice. The polypeptide samples were diluted to 1 μg/μL using the buffer, the sample volume was set to 5 μL.

There is total lack of the validation step of the obtained LC-MS data. The obtained results should be validated with Western blot for 4 to 5 chosen proteins to confirm the correctness of the method and results. Please attach the description of the validation experiment and the obtained results.

Response: Thank you for your advice. At present, there are few specific antibodies against yak-related proteins. At the same time, due to the impact of the epidemic, some reagents cannot be bought. Therefore, considering various factors, I am sorry that we could not do follow-up experiments.

2.11. Data Analysis

Lines 234-235: Unnecessary space

Response: Thank you for your advice. Unnecessary space has been removed.

Did the Authors base on the P-value or on the FDR? In this case the FDR values are more important and commonly used for the high throughput data sets obtained from the transcriptomic or proteomic experiments.

Response: Thank you for your advice. The purpose of this experiment was to identify the proteins interacting with TSSK4 protein. Only can be qualitative of the proteins with TSSK4 protein interaction, but the quantification of differential proteins cannot be carried out. Therefore, p-value and FDR need not be set.

3.1. Identification and subcellular localization of the alternative spliceosomes of the yak Tssk4

It would be very beneficial for the present studies to confirm the predicted localization of the spliceosomes with the use of cellular in situ hybridization experiments.

Response 1: Thank you for your advice. Since tssk4 gene has several alternative spliceosomes with small differences, it was not suitable for verification by TSSK4 protein antibody. In the later experiment, TSSK4 protein was co-expressed with the tag protein to verify its cell location.

3.3. Identification of the TSSK4 interacting proteins by GST pull-down combined with the LC-MS/MS mass spectrometry

What was the Fold Change (FC) and p-value cutoff level?

Response: Thank you for your advice. The purpose of this experiment was to identify the proteins interacting with TSSK4 protein. Only can be qualitative of the proteins with TSSK4 protein interaction, but the quantification of differential proteins cannot be carried out. Therefore, p-value and FC need not be set.

  • Discussion

The discussion is very descriptive and should be remodeled. I suggest the authors to focus more on the individual proteins revealed in the LC-MS analysis than on the GO results.

Response: Thank you for your advice. Enrichment analysis of proteins identified by LC-MS, focusing on the function of proteins identified by LC-MS. The content is as follows: The proteins identified by LC-MS/MS were enriched in the estrogen signaling pathway. Estrogen was first isolated nearly 90 years ago accounting for possible roles pertaining to female reproduction. With expanding research, estrogen has also been found to be produced in men. The estrogen receptors (ER) are abundantly expressed in the Mullerian and Wolfe ducts of the developing male reproductive tract and the urogenital sinus, as well as in the reproductive organs of the late fetuses and newborns derived from the Mullerian and Wolfe ducts. Cloning of human ER and de-velopment of ER antibodies has recognized the efferent tubule as the main target of male estrogen. The reproductive phenotypes of the Esr1 KO male rats are similar to that of the Esr1 KO mice, which include infertility, decreased testicular weight, and sperm in the epididymal tail. However, estrogen has been recently identified to play a role in spermatogenesis in the testes and epididymis and to maintain certain aspects of epididymal physiology. Estrogens, ER, and the whole estrogen sig-naling pathway are important for regulating the male fertility, development, and func-tion of the efferent ducts and prostate, and the flow of the adult sperm from the testis to the epididymis.

Line 374, after (2): The capital letter is unnecessary.

Response: Thank you for your advice. “(2) Since” has been replaced by “(2) since” in line 373.

Lines 386-387: The part of the discussion above, as well as the presented results did not support this thesis.

Response: Thank you for your advice. The sentences, this highlights that the TSSK protein family has a regulatory role in the cytoplasm during spermatogenesis after sexual maturity, have been deleted in line 386-387 of the article.

Lines 398-412: This part of discussion is only the description of the estrogen role in the male reproductive system and is not addressed to the obtained results and its potential impact on the physiology.

Response: Thank you for your advice. This part mainly introduces the Estrogen signaling pathway enriched by proteins interacting with TSSK4, and introduces the role of this pathway in spermatogenesis.

Sincerely,

Xian Guo

[email protected]

Reviewer 2 Report

This is an interesting and well-written manuscript focused to analyze the variable spliceosome of the TSSK4 gene in yak for understanding the regulatory function of TSSK4 spliceosome in yak testis development.

Introduction and Methods are appropriate to explain the research question, which allowed to find very interesting Results and to write a well-supported Discussion. However, there are some important major comments that should be considered:

  • “Simple Summary” section is missing.
  • “Abstract” section should describe briefly the main methods or treatment applied.
  • “Materials and Methods” did no include any procedure or software used for GO and KEGG enrichment analyses.
  • “Conclusions” include more results than final conclusions. Please only include one or two paragraphs in this section as mentioned in the “Instructions for Authors”.
  • “References” section must follow guidelines described in the “Instructions for Authors”. There are two errors in this section: (a) The name of the Journal is not abbreviated in most of the references, and (b) Only the first letter should be capitalized in the article name, which is written incorrectly in several references.
  • “Data Availability Statement” is not mentioned in the manuscript.

I also suggest considering next minor grammar comments in order to improve the quality of the manuscript:

  • Line 147: Replace “are” by “were”.
  • Line 157: Remove the period after the phrase “constant temperature”.
  • Line 160: Replace “qing ke biotechnology co., LTD.” by “Xi'an Qingke Biotechnology Co., Ltd.”
  • Line 174: Replace “the” by “The”.
  • Line 175: Replace the comma by a period after the phrase “for 2 h”.
  • Line 183: Replace the semicolon by a period.
  • Line 184: Remove the comma.
  • Line 185: Replace “at2500” by “at 2500”.
  • Lines 199 to 205: Very long sentence; please separate it in two sentences.
  • Lines 235 to 236: Join the two sentences.
  • Line 259: The next phrase “please refer to Table 2 for details” should be placed in round brackets.
  • Line 304: Replace “2C” by “2c”.
  • Lines 354 and 355: Remove the round brackets.
  • Line 373: Replace “the” by “The”.
  • Line 380: Replace the phrase “Studies by Li et al.” by “A previous study”.
  • Line 441: Replace the comma by a period after the phrase “molecular function”.
  • Line 464: Replace “genomics” by “Genomics”.

Author Response

Dear Reviewer:

Thank you for your time concerning our manuscript entitled “Identification of the Tssk4 alternative spliceosomes and analy-sis of the function of the TSSK4 protein in yaks” (written by Xingdong Wang et al.), which we wish to be considered for publication in “Animals”. The following is a point-by-point response to your comments.

“Simple Summary” section is missing.

Response: Thank you for your advice. The Simple Summary has been added to lines 13-18 of the article as requested. The details are as follows: Simple Summary: In mammals, the testis-specific serine/threonine kinase (TSSK) is essential for spermatogenesis and male fertility. This study aimed to analyze the possible alternative spliceosomes of the tssk4 gene in the yak testis tissue using PCR amplification and cloning tech-niques. A total of 6 alternative spliceosomes were obtained, of which there were only 2 alternative spliceosomes in the yak testis before sexual maturity and 4 alternative spliceosomes in the yak testis after sexual maturity.

“Abstract” section should describe briefly the main methods or treatment applied.

Response: Thank you for your advice. The main experimental methods have been added in the abstract of line 24-25 of the paper, as follows: The GST pull-down was used for pulling down the protein interacting with TSSK4, and then the protein interacting with TSSK4 was identified using the LC-MS/MS.

“Materials and Methods” did no include any procedure or software used for GO and KEGG enrichment analyses.

Response: Thank you for your advice. "The functional enrichment analysis of interacting genes with tssk4 was performed using the g: Profile online website (https://biit.cs.ut.ee/gprofiler/gost) and KOBAS online website (http://kobas.cbi.pku.edu.cn/)." has been added in the article in line 261-264.

“Conclusions” include more results than final conclusions. Please only include one or two paragraphs in this section as mentioned in the “Instructions for Authors”.

Response: Thank you for your advice. The conclusions have been refined as requested and detailed as follows: This experiment identified six alternative spliceosomes of the tssk4 gene at four stages of yak using PCR amplification and cloning technology. Among them, only two alternative spliceosomes were found in the yak before sexual maturity (6 months old and 18 months old). After sexual maturity (30 months old and 5 years old), four alternative spliceosomes were amplified in the yak.

“References” section must follow guidelines described in the “Instructions for Authors”. There are two errors in this section: (a) The name of the Journal is not abbreviated in most of the references, and (b) Only the first letter should be capitalized in the article name, which is written incorrectly in several references.

Response: Thank you for your advice. The reference format in the article has been revised as required.

References

  1. Das, P.P.; Krishnan, G.; Doley, J.; Biswas, T.K.; Paul, V.; Chakravarty, P.; Deb, S.M.; Das, P.J. Identification and expression profiling of MSY genes of yak for bull fertility. J Genet 2019, 98, 1-10.
  2. Hui, W.; Chai, Z.; Dan, H.; Ji, Q.; Zhong, J. A global analysis of CNVs in diverse yak populations using whole-genome resequencing. BMC Genomics 2019, 20, 61.
  3. Yang, C.; Ahmad, A.A.; Bao, P.J.; Guo, X.; Ding, X.Z. Increasing dietary energy level improves growth performance and lipid metabolism through up-regulating lipogenic gene expression in yak (Bos grunniens). Anim Feed Sci Technol 2020, 263, 114455.
  4. Wiener, G.; Han, J.; Long, R. The yak. Rap Publication 2003, 44, 57-58.
  5. Wang, X.; Pei, J.; Bao, P.; Cao, M.; Guo, S.; Song, R.; Song, W.; Liang, C.; Yan, P.; Guo, X. Mitogenomic diversity and phylogeny analysis of yak (Bos grunniens). BMC Genomics 2021, 22, 325.
  6. Li, Z.; Jiang, M. Metabolomic profiles in yak mammary gland tissue during the lactation cycle. PLoS One 2019, 14, e0219220.
  7. Jiang, M.; Lee, J.; Bionaz, M.; Deng, X.; Wang, Y. Evaluation of suitable internal control genes for RT-qPCR in yak mammary tissue during the lactation cycle. PLoS One 2016, 11, e0147705.
  8. Zhou, X.; Wu, X.; Chu, M.; Liang, C.; Ding, X.; Pei, J.; Xiong, L.; Bao, P.; Guo, X.; Yan, P. Validation of suitable reference genes for gene expression studies on yak testis development. Animals 2020, 10, 182.
  9. Wang, P.; Huo, H.L.; Wang, S.Y.; Miao, Y.W.; Xiao, H. Cloning, sequence characterization, and expression patterns of members of the porcine TSSK family. Genetics & Molecular Research Gmr 2015, 14, 14908.
  10. Wang, Z.; Cole, P.A. Catalytic mechanisms and regulation of protein kinases. Methods Enzymol 2014, 548, 1-21.
  11. Wang, X.; Wei, Y.; Fu, G.; Li, H.; Saiyin, H.; Lin, G.; Wang, Z.; Chen, S.; Yu, L. Tssk4 is essential for maintaining the structural integrity of sperm flagellum. Mol Hum Reprod 2015, 21, 136-145.
  12. Su, D.; Zhang, W.; Yang, Y.; Deng, Y.; Ma, Y.; Song, H.; Zhang, S. Mutation screening and association study of the TSSK4 gene in chinese infertile men with impaired spermatogenesis. J Androl 2008, 29, 374-378.
  13. Shang, P.; Baarends, W.M.; Hoogerbrugge, J.; Ooms, M.P.; Cappellen, W.V.; Jong, A.D.; Dohle, G.R.; Eenennaam, H.V.; Gossen, J.A.; Grootegoed, J.A. Functional transformation of the chromatoid body in mouse spermatids requires testis-specific serine/threonine kinases. J Cell Sci 2010, 123, 331-339.
  14. Jha, K.; Coleman, A.; Wong, L.; Salicioni, A.; Howcroft, E.; Johnson, G. Heat shock protein 90 functions to stabilize and activate the testis-specific serine/threonine kinases, a family of kinases essential for male fertility. J Biol Chem 2013, 288, 16308-16320.
  15. Wang, X.; Li, H.; Fu, G.; Wang, Y.; Du, S.; Yu, L.; Wei, Y.; Chen, S. Testis-specific serine/threonine protein kinase 4 (Tssk4) phosphorylates Odf2 at Ser-76. Sci Rep 2016, 6, 22861.
  16. Chen, X.; Lin, G.; Wei, Y.; Hexige, S.; Niu, Y.; Liu, L.; Yang, C.; Yu, L. TSSK5, a novel member of the testis-specific serine/threonine kinase family, phosphorylates CREB at Ser-133, and stimulates the CRE/CREB responsive pathway. Biochem Biophys Res Commun 2005, 333, 742-749.
  17. Kueng, P.; Nikolova, Z.; Djonov, V.; Hemphill, A.; Rohrbach, V.; Boehlen, D.; Zuercher, G.; Andres, A.; Ziemiecki, A. A novel family of serine/threonine kinases participating in spermiogenesis. J Cell Biol 1997, 139, 1851-1859.
  18. Wei, Y.; Wang, X.; Fu, G.; Yu, L. Testis specific serine/threonine kinase 4 (Tssk4) maintains its kinase activity by phosphorylating itself at Thr-197. Mol Biol Rep 2013, 40, 439-447.
  19. Salicioni, A.M.; Gervasi, M.G.; Julian, S.; Tourzani, D.A.; Saman, N.; Caraballo, D.A.; Visconti, P.E. Testis-specific serine kinase protein family in male fertility and as targets for non-hormonal male contraception†. Biol Reprod 2020, 103, 264-274.
  20. Wang, X.; Wei, Y.; Fu, G.; Yu, L. Testis specific serine/threonine protein kinase 4 (TSSK4) leads to cell apoptosis relying on its kinase activity. J Huazhong Univ Sci Technolog Med Sci 2015, 35, 235-240.
  21. Sun, X.; Tian, Y.; Wang, J.; Sun, Z.; Zhu, Y. Genome-wide analysis reveals the association between alternative splicing and DNA methylation across human solid tumors. BMC Med Genomics 2020, 13, 4.
  22. Mthembu, N.N.; Zukile, M.; Rodney, H.; Zodwa, D. Abnormalities in alternative splicing of angiogenesis-related genes and their role in HIV-related cancers. HIV AIDS (Auckl) 2017, 9, 77-93.
  23. Zhao, S. Alternative splicing, RNA-seq and drug discovery. Drug Discov Today 2019, 24, 1258-1267.
  24. Bowler, E.; Oltean, S. Alternative splicing in angiogenesis. Int J Mol Sci 2019, 20, 2067.
  25. Matlin, A.; Clark, F.; Smith, C. Understanding alternative splicing: towards a cellular code. Nat Rev Mol Cell Bio 2005, 6, 386-398.
  26. Choi, N.; Liu, Y.; Oh, J.; Ha, J.; Zheng, X.; Shen, H. U2AF65-dependent SF3B1 function in SMN alternative splicing. Cells 2020, 9, 2647.
  27. Du, J.X.; Zhu, G.Q.; Cai, J.L.; Wang, B.; Dai, Z. Splicing factors: Insights into their regulatory network in alternative splicing in cancer. Cancer Lett 2020, 501, 83-104.
  28. Ule, J.; Blencowe, B.J. Alternative splicing regulatory networks: functions, mechanisms, and evolution. Mol Cell 2019, 76, 329-345.
  29. Jin, L.; Chen, Y.; Crossman, D.K.; Datta, A.; Vu, T.; Mobley, J.A.; Basu, M.K.; Scarduzio, M.; Wang, H.; Chang, C. STRAP regulates alternative splicing fidelity during lineage commitment of mouse embryonic stem cells. Nat Commnu 2020, 11, 5941.
  30. Biamonti, G.; Catillo, M.; Pignataro, D.; Montecucco, A.; Ghigna, C. The alternative splicing side of cancer. Semin cell dev biol 2014, 32, 30-36.
  31. Paronetto, M.P.; Passacantilli, I.; Sette, C. Alternative splicing and cell survival: from tissue homeostasis to disease. Cell Death Differ 2016, 23, 1919-1929.
  32. Li, X.; Li, Y.; Song, W.; Xie, D.; Li, Y. cDNA Cloning, expression and bioinformatical analysis of tssk genes in tree shrews. Comput Biol Chem 2021, 92, 107474.
  33. Salicioni, A.M. Expression and localization of five members of the testis-specific serine kinase (Tssk) family in mouse and human sperm and testis. Mol Hum Reprod 2011, 17, 42.
  34. Chen, Y.M.; Chen, C.; Riley, D.J.; Allred, D.C.; Lee, W.H. Aberrant subcellular localization of BRCA1 in breast cancer. Science 1995, 270, 789-791.
  35. Hung, M.C.; Link, W. Protein localization in disease and therapy. J Cell Sci 2011, 124, 3381-3392.
  36. Korobova, F.; Ramabhadran, V.; Higgs, H.N. An actin-dependent step in mitochondrial fission mediated by the ER-associated formin INF2. Science 2013, 339, 464.
  37. Loveland, K.; Hayes, T.; Meinhardt, A.; Zlatic, K.; Parvinen, M.; de Kretser, D.; McFarlane, J. Microtubule-associated protein-2 in the rat testis: a novel site of expression. Biol reprod 1996, 54, 896-904.
  38. Berthrong, M.; Goodwin, W.E.; Scott, W.W. Estrogen production by the testis. J Clin Endocr Metab 1949, 9, 579.
  39. Stumpf, W.E.; Narbaitz, R.; Sar, M. Estrogen receptors in the fetal mouse. J Steroid Biochem 1980, 12, 55-64.
  40. Walter, P.; Green, S.; Greene, G.; Krust, A.; Bornert, J.; Jeltsch, J.; Staub, A.; Jensen, E.; Scrace, G.; Waterfield, M. Cloning of the human estrogen receptor cDNA. P Natl Acad Sci USA 1985, 82, 7889-7893.
  41. Greene, G.; Closs, L.; Fleming, H.; DeSombre, E.; Jensen, E. Antibodies to estrogen receptor: immunochemical similarity of estrophilin from various mammalian species. P Natl Acad Sci USA 1977, 74, 3681-3685.
  42. Cooke, P.; Young, P.; Hess, R.; Cunha, G. Estrogen receptor expression in developing epididymis, efferent ductules, and other male reproductive organs. Endocrinology 1991, 128, 2874-2879.
  43. Rumi, M.; Dhakal, P.; Kubota, K.; Chakraborty, D.; Lei, T.; Larson, M.; Wolfe, M.; Roby, K.; Vivian, J.; Soares, M. Generation of ESR1-knockout rats using zinc finger nuclease-mediated genome editing. Endocrinology 2014, 155, 1991-1999.
  44. Cooke, P.S.; Nanjappa, M.K.; Ko, C.M.; Prins, G.S.; Hess, R.A. Estrogens in male physiology. Physiol Rev 2017, 97, 995-1043.
  45. Dumasia, K.; Kumar, A.; Deshpande, S.; Balasinor, N.H. Estrogen, through estrogen receptor 1, regulates histone modifications and chromatin remodeling during spermatogenesis in adult rats. Epigenetics 2017, 12, 953-963.
  46. Dumasia, K.; Kumar, A.; Deshpande, S.; Sonawane, S.; Balasinor, N.H. Differential roles of estrogen receptors, ESR1 and ESR2, in adult rat spermatogenesis. Mol Cellular Endocrinol 2016, 428, 89-100.
  47. Cavalcanti, F.N.; Lucas, T.; Lazari, M.; Porto, C.S. Estrogen receptor ESR1 mediates activation of ERK1/2, CREB, and ELK1 in the corpus of the epididymis. J Mol Endocrinol 2015, 54, 339-349.
  48. Joseph, A.; Hess, R.A.; Schaeffer, D.J.; Ko, C.; Hudgin-Spivey, S.; Chambon, P.; Shur, B.D. Absence of estrogen receptor alpha leads to physiological alterations in the mouse epididymis and consequent defects in sperm function. Biol Reprod 2010, 82, 948-957.
  49. Hess, R.A.; Cooke, P.S. Estrogen in the male: a historical perspective. Biol Reprod 2018, 99, 27-44.

“Data Availability Statement” is not mentioned in the manuscript.

Response: Thank you for your advice. Data Availability Statement has been added in Lines 444-446. Details are as follows: The datasets generated and analyzed during the current study are available in the GenBank repository, accession number: MT950338 -- MT950343. The data sup-porting the conclusions of this study are available within the supplementary table.

I also suggest considering next minor grammar comments in order to improve the quality of the manuscript:

Response: Thank you for your advice. We have checked and revised the grammar of the article.

  • Line 147: Replace “are” by “were”.
  • Line 157: Remove the period after the phrase “constant temperature”.
  • Line 160: Replace “qing ke biotechnology co., LTD.” by “Xi'an Qingke Biotechnology Co., Ltd.”
  • Line 174: Replace “the” by “The”.
  • Line 175: Replace the comma by a period after the phrase “for 2 h”.
  • Line 183: Replace the semicolon by a period.
  • Line 184: Remove the comma.
  • Line 185: Replace “at2500” by “at 2500”.
  • Lines 199 to 205: Very long sentence; please separate it in two sentences.
  • Lines 235 to 236: Join the two sentences.
  • Line 259: The next phrase “please refer to Table 2 for details” should be placed in round brackets.
  • Line 304: Replace “2C” by “2c”.
  • Lines 354 and 355: Remove the round brackets.
  • Line 373: Replace “the” by “The”.
  • Line 380: Replace the phrase “Studies by Li et al.” by “A previous study”.
  • Line 441: Replace the comma by a period after the phrase “molecular function”.
  • Line 464: Replace “genomics” by “Genomics”.

Response: Thank you for your advice. The above details have been revised as required.

Sincerely,

Xian Guo

[email protected]

Reviewer 3 Report

I reviewed the manuscript entitled "Identification of the Tssk4 alternative spliceosomes and analysis of the function of the TSSK4 protein in yaks", and I found it relevant to the scientific literature.

The introduction is described sufficiently and introduces the reader to the topic undertaken by the researchers.  

Material and methods are correctly and comprehensively described with the right amount of details., which is necessary for this article. 

The results are presented logically and allow the reader to follow the analysis course. The tables and figures are informative. 

In the discussion, the authors discussed the research results in relation to the literature and touched upon the most important problems arising from the research. The discussion is written in a way that is understandable to the reader.

 In my opinion, the work is written in the correct language and understandable to the reader. I have to admit that this is rare, but I have no significant comments on this article.

Author Response

Thank you!

Round 2

Reviewer 1 Report

 The revised version of Manuscript is suitable for publication, however, it would be benefitial to rethink some parts of the Discussion section to make the Manuscript more attractive.

Reviewer 2 Report

The Manuscript has improved significantly. I consider that the revision version of the Manuscript is now suitable for publication. I only suggest to check Journal name abbreviation in line 488.